# The Dual Role of Oxidants in Male (In)fertility: Every ROSe Has a Thorn

**DOI:** 10.3390/ijms24054994

**Published:** 2023-03-05

**Authors:** Antonio Mancini, Alessandro Oliva, Edoardo Vergani, Roberto Festa, Andrea Silvestrini

**Affiliations:** 1Dipartimento di Medicina e Chirurgia Traslazionale, Università Cattolica del Sacro Cuore, 00168 Rome, Italy; 2Fondazione Policlinico Universitario Agostino Gemelli IRCCS, 00168 Rome, Italy; 3Dipartimento di Scienze Biotecnologiche di Base, Cliniche Intensivologiche e Perioperatorie, Università Cattolica del Sacro Cuore, 00168 Rome, Italy

**Keywords:** male infertility, spermatozoa, oxidative stress, capacitation, antioxidants

## Abstract

The role of oxidative stress (OS) in male infertility as a primary etiology and/or concomitant cause in other situations, such as inflammation, varicocele and gonadotoxin effects, is well documented. While reactive oxygen species (ROS) are implicated in many important roles, from spermatogenesis to fertilization, epigenetic mechanisms which are transmissible to offspring have also recently been described. The present review is focused on the dual aspects of ROS, which are regulated by a delicate equilibrium with antioxidants due to the special frailty of spermatozoa, in continuum from physiological condition to OS. When the ROS production is excessive, OS ensues and is amplified by a chain of events leading to damage of lipids, proteins and DNA, ultimately causing infertility and/or precocious pregnancy termination. After a description of positive ROS actions and of vulnerability of spermatozoa due to specific maturative and structural characteristics, we linger on the total antioxidant capacity (TAC) of seminal plasma, which is a measure of non-enzymatic non-proteic antioxidants, due to its importance as a biomarker of the redox status of semen; the therapeutic implications of these mechanism play a key role in the personalized approach to male infertility.

## 1. Introduction

Male infertility and oxidative stress (OS) are closely related, as reported in several papers [1,2,3,4,5]. OS is associated with many risk factors for infertility, such as varicocele, inflammation, metabolic alterations, endogenous or exogenous toxins and radiofrequency [6]; however, it can also be observed as the only factor that causes unexplained infertility. Accordingly, the term MOSI (male-oxidative-stress-related infertility) indicates such a situation [7]. Since this mechanism is frequent, the administration of antioxidants has prevailed in clinical practice, usually without a previous evaluation of the effective presence of OS damage. Such an approach may be inappropriate, since the metaphor of a double-edged sword [6,8] should be considered when evaluating the physiological role and/or pathological effects of oxidants, often reported as reactive oxygen species (ROS).

The knowledge of the dual role of ROS has deep roots. Three phases of this topic can be described [9]. The first phase began in 1943, when McLeod [10] showed an impairment in spermatozoa motility when incubating human sperm in conditions of high oxygen tension. This observation was pioneering, since it was generalized to human physiology; in 1954, a relationship was established with oxygen toxicity and its reduced forms [11], followed by the discovery of paramagnetic resonance signals related to free radicals in lyophilized biological materials in the same year [12], and in 1956 the hypothesis of ageing based on increased radical production [13].

The second phase began with the discovery of superoxide dismutase (SOD) in 1969, underling the key role of free radicals in living systems [14]. Finally, in the third phase, evidence showing ROS-stimulating action on the guanylate-cyclase and therefore on the production of cGMP in 1977 [15] pointed out the concept of the beneficial effect of ROS at low/moderate concentrations, while deleterious effects appear at higher levels or when the buffering effect of the antioxidant is overwhelmed.

In particular, ROS are involved in cytokine and growth-factor signaling, the activation of non-receptor tyrosine kinases, several serine/threonine kinases, the inactivation of protein tyrosine phosphatases and the modulation of nuclear transcription factors [9]. H_2_O_2_ is an important example of this role, despite its very low concentration and very slow reactivity with redox-regulated proteins, in comparison with thiol peroxidases, which inactivate it [16]. This paradox has been clarified, suggesting that the thiol peroxidase peroxiredoxin-2 (Prx2), a very highly reactive cellular protein, can act as an H_2_O_2_ receptor and amplify the transmission, forming a redox relay with the transcription factor STAT3 [17]. The peroxiredoxins (which will be discussed below in relation to infertility) have a complex modulation, which can shift from signaling to scavenging properties.

These cellular mechanisms are common to multiple physiological events, including cardiomyocyte cell functioning [18], regulation of vascular tone [19], sensing of changes in oxygen concentrations [20], cell adhesion [21], immune response [22] and, ultimately, in cell lifecycle, since they are involved in apoptosis/survival [23]. Therefore, it is not surprising that the control of ROS production and its consequences is crucial for a well-differentiated cell, such as spermatozoa, which need a long period of maturation in the testis and epididymis, and structural arrangements allowing them to perform a safe “travel” outside the body, in the female genital tract, and ultimately to fertilize the oocyte.

Due to these complex mechanisms, a clear complete picture still has not been obtained. The present review therefore aims to synthetize the following topics: the physiological positive role of ROS, the unique structure of spermatozoa leading to vulnerability to OS and, finally, our experience with the evaluation of total antioxidant capacity (TAC) in seminal plasma as biological marker of male infertility.

## 2. Positive Effects of ROS

Surprisingly enough, cells such as spermatozoa, rich in substrates suitable for pre-venting oxidative stress and its perpetration, generate huge amounts of ROS. It is universally known that oxidative stress may impair the normal functioning of spermatozoa by reducing their motility and damaging their DNA through the direct and indirect oxidation of membrane lipids and nucleic acids. Despite this, spermatozoa are dependent from oxidative phosphorylation, which notably occurs in mitochondria and generates ROS because of electron leakage during the process. Moreover, spermatozoa usually have between 50 and 75 mitochondria, entirely located between the head and the flagellum. This leads to a significant production of ROS, among which the main species are represented by hydroxyl radicals (^•^OH), superoxide anion (^•^O_2_^−^), hydrogen peroxide (H_2_O_2_) and nitric oxide (NO) [24]. The central role of mitochondria is documented by the positive correlation between ROS generation by these organelles and sperm motility, with a consequent positive effect on pregnancy rate in vivo [25]. The compartmentalization of mitochondria in the mid-piece results in a physical separation from nuclear DNA, thus preventing possible damage in case of ROS overproduction.

To balance the effect of ROS, spermatozoa are endowed with several antioxidant enzymes, albeit at a low concentration. Among these, superoxide dismutase, catalase and glutathione peroxidase are noteworthy; the former is involved in superoxide anion reduction while the latter two are mostly involved in hydrogen peroxide reduction [26]. The previously cited enzymes represent the intracellular side of antioxidant production. Apart from this, it is notable that seminal fluid itself is rich in antioxidants as well, both enzymatic and non-enzymatic; the total antioxidant power of seminal plasma was estimated to be 10 times higher than that of blood [27]. Among the antioxidants of the seminal plasma, some are worthy of mention: vitamin C, uric acid, glutathione, taurine and hypotaurine. A large part of the existing literature focuses on analyzing the effects of the exogenous administration of antioxidants, such as coenzyme Q10, L-acetyl-Carnitine, vitamin C and Zinc. The general consensus is that antioxidants counteract oxidative stress in semen and grant better results in terms of fertility; however, uncontrolled or excessive treatment with antioxidants may lead to an impairment in sperm function, which delineates a clinical entity known as reductive stress [28]. This observation suggests that sperm function rests on a thin edge between oxidation and reduction, and that spermatozoa are damaged by large amounts of oxidants, but also cannot properly function without it, which brings us to the original question: why do spermatozoa generate ROS at all?

ROS have an established role as signal transductors in several cell types, but in spermatozoa they mediate the processes by which spermatozoa become capable of fertilizing: capacitation, hyperactivation and acrosome reaction. Moreover, ROS take part in the binding process with the oocyte, play a crucial role in compacting chromatin, and can modulate steroidogenesis.

Capacitation is triggered in the female genital tract and consists of a series of metabolic and structural changes in spermatozoa that make them ready for fertilization. In short, superficial molecules present in the female genital tract strip “decapacitating factors” from the membrane of sperm cells, leading to rearrangement of membrane cholesterol and sphingolipids, inactivation of ATP-dependent Ca^2+^ regulatory channel (PMCA) and alkalinization of intracellular pH [29]. Moreover, Ca^2+^ is capable of activating a cytosolic adenylate cyclase, which in turn activates protein kinase A (PKA) through cAMP. The PKA then phosphorylates residues of Thr, Tyr and Ser, which activate a protein tyrosine kinase (PTK) and thus allow phosphorylation of more Tyr residues, signaling the activation of the sperm cell. Other signaling cascades, which are modulated by these phosphorylations, include ERK- and SRC (Rous Sarcoma oncogene)-mediated pathways [30,31]. The other role of PKA is to stimulate the NADPH oxidase to produce ROS; on the other hand, ROS are able to activate the cytosolic adenylate cyclase, thus amplifying this process, and by the inhibition of the action of phosphatases, limit its termination [32]. As pointed out by de Lamirande et al. [33], reduction of ROS does not inhibit capacitation, but can hinder it to the point that the subsequent step, hyperactivation, cannot start.

As a consequence of capacitation, spermatozoa have an alkaline intracellular pH and a hyperpolarized membrane, which leads to a rise in Ca^2+^ dismissal from the endoplasmic reticulum and, subsequently, an increase of Ca^2+^ admission from the extracellular side. This leads to enhanced cAMP generation from the intracellular adenylate cyclase ADCY10, which activates PKA and thus permits the phosphorylation of Ser, Thr and Tyr residues of dynein and axokinin in the axonema, consequently placing the flagellum in a hyperactivated state. Both cAMP and Ca^2+^ increase the rate and width of oscillations of the flagellum in a process defined as hyperactivation, which is crucial for spermatozoa to progress through the oviduct [34]. Similarly to what happens during capacitation, ROS have the ability to activate ADCY10 and thus to enhance phosphorylation. Interestingly, the supplementation of the SOD enzyme in semen prevents hyperactivation, thus highlighting the role of ROS in this process [35]. Despite this review being focused on non-enzymatic antioxidants, it must be remarked that SOD is an important antioxidant defense in protecting spermatozoa. It has been shown that SOD activity in seminal plasma had a positive correlation with sperm concentration and total motility and a negative one with DNA fragmentation; interestingly, SOD levels can be influenced by diet but also by genetic polymorphism, since the variant genotype SOD2 rs4880 was associated with low SOD activity [36].

When spermatozoa encounter the cumulus surrounding the oocyte, their contact with the zona pellucida activates a transformation process known as acrosome reaction. In humans, contact with zona pellucida protein ZP3 seems to activate Ca^2+^ channels, which generate a transient Ca^2+^ increase in the sperm cell, which in turn allow for PI3K activation, generation of PIP3, activation of serine proteases PKB and PKCζ and ultimately exocytosis of molecules such as acrosin [37]. Acrosin is a serine protease which, among other things, helps to digest the membrane of spermatozoa, allowing the binding with the zona pellucida [32]. De Lamirande et al. [33] demonstrated that acrosomal reaction could be induced only in capacitated spermatozoa, and that it could be inhibited by adding either SOD or catalase, indicating that both ^•^O_2_^−^ and H_2_O_2_, respectively, have a role in acrosomal reaction. Moreover, the authors speculated on the possible underlying mechanisms of ROS participation in the process, such as lipid peroxidation of the membrane and tyrosine phosphorylation of membrane proteins. Regarding the latter mechanism, nitric oxide (NO) has been shown to participate in the process as well, by inducing protein phosphorylation through cGMP synthesis and subsequent protein kinase activation [38].

After acrosome reaction, a spermatozoon binds to the zona pellucida of the oocyte, allowing genetic material to be incorporated alongside that of the oocyte. The previous process requires the activation of Phospholipase A2 (PLA2), which rearranges membrane lipids and permits fusion between the membranes. Goldman et al. [39] showed that oxidants such as ^•^O_2_^−^ and H_2_O_2_ are capable of activating PKC, which in turn activates PLA2, thus participating in the sperm–oocyte binding. It is interesting to note that oocyte generates ROS as well after the binding of the spermatozoon to the zona pellucida. ROS production induces the phenomenon known as “zona hardening” in order to avoid polyspermy [40]. Finally, spermatozoa provide components which induce oocyte activation, such as phospholipase C-zeta [41] and RNAs [42,43], which can play a role in epigenetic mechanisms in early embryo development [43].

Chromatin condensation is an important process spermatozoa undergo to protect DNA from oxidative damage. It is known that an excessive production of ROS overcomes antioxidant systems and can generate the adduct molecule 8-hydroxy-deoxy-guanosine (8OHdG), which ultimately leads to the loss of the affected base and the subsequent generation of a strand break. However, a lower amount of ROS activates a sperm-specific phospholipid-hydroperoxide glutathione peroxidase. This enzyme catalyzes the protamination of bases to generate intramolecular disulfide bridges, which renders chromatin more compact and protects DNA from external insults [27], as detailed below; this is a crucial point due to the “journey” sperm cells embark on.

A decisive, yet often overlooked, role played by ROS in spermatozoa is the regulation of apoptosis. In fact, the regulation of apoptosis is fundamental in maintaining an adequate germ/Sertoli cell ratio during spermatogenesis, while at the same time eliminating defective sperm cells, determining an overall increase of fertility [44]. During cell life, apoptosis may occur via the intrinsic or the extrinsic pathway. Interestingly, spermatozoa seem to undergo apoptosis almost exclusively via the former [45]. Normally, ROS damage is responsible for lipid peroxidation of the mitochondria, membrane damage and activation of the caspase cascade [42]. In spermatozoa, however, mitochondria are strictly separated from the nucleus, and thus caspases cannot move to the head of the spermatozoon. The only molecule that can do so is H_2_O_2_, produced in the mitochondria and released from them during apoptosis. Hydrogen peroxide can then produce molecules such as acrolein and 4-HNE, which bind DNA, leading to its damage and, ultimately, cell death via apoptosis [46]. ROS appear to regulate this process through phosphorylation and de-phosphorylation of kinases such as PI3K, which in turn activates a cascade that leads to the phosphorylation of BCL2-associated death promoter (BAD), an important promoter of the caspase cascade, thus inactivating it [46]. The importance of phosphorylation and oxidation processes in orchestrating apoptosis has led to speculation about the nature of spermatozoa maturation. It has been hypothesized that capacitation is the same event that, through overproduction of ROS, leads to induction of apoptosis. After ejaculation, spermatozoa initiate capacitation shortly before their approach to the site of fertilization. If they do not achieve fertilization, ROS produced during capacitation ultimately overwhelm the defenses of the spermatozoa, leading to its apoptosis. Thus, paradoxically, the spermatozoa, the cells responsible for creating life, appear to actually be designed for death [47]. A summary of the main features is schematized in Figure 1.

## 3. Vulnerability of Spermatozoa

Spermatozoa are particularly vulnerable to oxidative damage due to multiple intrinsic structural characteristics of these specialized cells, but also due to the complex mechanism of differentiation (i.e., spermatogenesis) and maturation required to reach a full fertilizing capacity. During spermatogenesis, sperm chromatin undergoes many changes that lead to a tighter quaternary structure. After meiosis, histones are replaced with transition proteins, which in turn will be replaced with protamines type 1 and type 2. This allows the nucleus to be 6–7 times smaller than other cells’ nuclei. However, 5–10% of the sperm DNA are free from the binding with protamines. These portions are distributed in regions called “solenoids”, where they are bound with paternal nucleosome. These regions are more accessible to ROS due to their more relaxed structure, thus leading to DNA damage and possible fragmentation [48]. This represents the first step according to the “two-steps” hypothesis proposed by Aitken [49]. As for the second step, oxidative damage of DNA results in the generation of 8-desoxyguanosine (8OHdG) adducts. Spermatozoa are endowed with a limited base excision repair (BER) system. Although they possess an 8-oxoguanine glycosylase 1 (OGG1) enzyme, they lack the subsequent enzymes in the pathway, apurinic endonuclease 1 (APE1) and X-ray repair cross-complementing protein 1 (XRCC1). The former’s role is to cleave phosphate groups at the 3′ and 5′ of the baseless site, while the latter interacts with many enzymes in order to restore the normal DNA structure [50]. Thus, when an oxidative insult generates 8OHdG, OGG1 cleaves the adduct from DNA, generating an abasic site which cannot be further repaired by the spermatozoa, which either becomes totally dependent from the repairing system of the oocyte [44] or undergoes DNA fragmentation [51]. At the end of spermatogenesis, spermatozoa have expelled a large part of their cytoplasm. This means that cytosolic antioxidant systems are lacking as a result. Among the main antioxidants, catalase, superoxide dismutase (SOD) and glutathione peroxidase are noteworthy [49], but phospholipase A2 is also critical for its role in cleaving oxidized fatty acids, leading to their presentation to glutathione peroxidase. Finally, peroxiredoxins play an important role as well [52]. Peroxiredoxins are sulfydryl-dependent, selenium- and heme-free peroxidases, with high evolutionary conservation [53]. They play a role of importance in the oxidative milieu of the spermatozoa as they act as antioxidants, reducing highly reactive molecules such as H_2_O_2_ and in turn oxidizing thiol compounds. [52]. Thus, peroxiredoxins are an active part of H_2_O_2_ signaling. In particular, peroxiredoxin-2 (Prx2) has been found to transfer oxidative equivalents to compounds, coupled with scaffold protein STAT3 [17] and guided specifically to thiols by chaperone annexin A2 [54]. It has been shown that inhibitors of peroxiredoxins prevent capacitation in humans [55]. Oxidative stress may impair glutathione peroxidase activity also by inhibiting glucose-6-phostphate dehydrogenase, thus reducing the production of NADPH which is necessary to reduce oxidized glutathione [51]. On the other hand, excess of cytoplasm could be a factor of vulnerability as well, since its retention due to an altered maturation enhances enzymes related to ROS production, such as creatine kinase and glucose-6-phosphate dehydrogenase [56].

Spermatozoa have a plasma membrane rich in polyunsaturated fatty acids. These molecules are characterized by the presence of double bonds in *cis* configuration separated by methylene groups. Being close to a methylene group renders the bonds particularly vulnerable to oxidation. Lipid peroxidation (LPO) generates lipid radicals that propagate oxidation across the membrane. The process leads to the generation of by-products such as malondialdehyde, 4hydroxynonenal (4HNE) and acrolein. Malondialdehyde is mutagenic, but can also form DNA adducts, leading to mutations in tumor suppressor genes [44], while 4NHE and acrolein exert mainly a direct genotoxic effect on DNA [51] while also increasing ROS production in mitochondria [50]. Recently, an interesting proteomic analysis showed that the main targets of 4-HNE are components of the mitochondrial respiratory chain [52].

While spermatozoa mainly rely on glycolysis for their energy catabolism, oxidative phosphorylation (OXPHOS) in the mitochondria is critical in supplying the spermatozoa with sufficient energy for its movements. Interestingly, OXPHOS is one of the main sources of OS in physiological conditions; nevertheless, they are themselves particularly prone to oxidative damage, which leads to exacerbated ROS generation. Firstly, ROS production initiates LPO in the mitochondrial membrane, leading to a dramatic surge in ROS generation through the disruption of mitochondrial membrane potential [57] and the dysfunction of electron transport complexes I and III [58]. However, it has also been suggested that activation of ROS generation at Complex III stimulated a rapid release of H_2_O_2_ into the extracellular space, without impact on peroxidative damage. On the contrary, the ROS generation by Complex I on the matrix side of the inner membrane sustained damage to the midpiece and deterioration of sperm motility. The previous mechanism can be counteracted by the presence of α-tocopherol [59]. Under this profile, it has been shown that catechol estrogens induce ROS generation in this matrix with detrimental effects [60]; this does not apply to 17β-estradiol, since it ameliorates mitochondrial respiration, augmenting complex IV activity [61]. Another distinctive of mitochondria in spermatozoa is the dynamic response to variation in energy requirements, mediated by large dynamin GTPases [62]. Some mitochondrial proteins seem to be associated with ROS regulation and ultimately on sperm quality: they include the uncoupling protein 2 (UCP2) and the protein DJ-1 (a product of the Parkinson’s-disease-related protein 7) which can promote a ROS removal role, and prohibitin, a chaperone protein of the inner mitochondrial membrane, inversely correlated with ROS generation [63]. Moreover, mitochondrial DNA is particularly susceptible to oxidative damage since it lacks both protection granted by histones and nucleotide-excision repair pathways. Damaged mitochondria render the spermatozoon unable to undergo apoptosis, which leads to the permanence of damaged and ROS-producing spermatozoa in the ejaculate, further increasing OS. Finally, mutations in mtDNA induced by ROS worsen electron leakage [44]. Accordingly, in spermatozoa, mtDNA appeared to be slightly more susceptible to DNA damage than nuclear DNA compared to other cell types and tissues [64]. In summary, mitochondria are critical for both physiology and pathological derangement, leading to infertility [65].

Paradoxically, even though spermatozoa present all these factors of vulnerability to oxidative damage, OS is nevertheless necessary and cannot be fully depleted from them. On the contrary, spermatozoa are “professional generators of ROS” [46], which is itself one of their main factors of vulnerability to OS. As stated before, oxido-reductase reactions happening in the mitochondria appear to be the main source of ROS in spermatozoa [66]. In addition to that, other systems are involved in the process. NADPH oxidases (NOX), and in particular NOX5, generate ^•^O_2_^−^ and are among the enzymes involved in capacitation, thus regulating sperm function [52]. As a note of interest, a regulatory role has been proposed for NOXs during spermatogonial cell proliferation [67]. L-amino acid oxidase was the first enzyme that was ever shown to generate ROS in mammalian spermatozoa and its relevance is thought to be related not only to capacitation, but to acrosome reaction as well [52]. Finally, lipoxygenases, and in particular 15-Lipoxygenase, are involved in the metabolism of PUFAs. The experimental inactivation of this lipoxygenase has been shown to attenuate ROS production, suggesting that lipoxygenases may be one of the main ROS producers in spermatozoa [47].

Notably, spermatozoa are subject not only to the ROS they produce, but also to those that are present in seminal plasma. The concentration of these molecules depends on endogenous (leukocytes, immature spermatozoa, varicocele) and exogenous factors (smoking, drinking, radiation, heat, high caloric intake) [51,66,68]. Their presence in seminal plasma correlates directly with alterations in semen quality, characteristically determining asthenozoospermia and augmented viscosity of semen [69]. Thus determining their presence in seminal plasma may be crucial, as they are a reversable cause of infertility.

The determination of ROS is made complex due to their very short half-lives. The measure of ROS is part of the advanced examination of semen in the Manual of the WHO [70]. They can be measured by chemiluminescent assay, using luminol as reactive, but a luminometer is not available in all laboratories. Recently, a direct evaluation system named MiOXSYS was proposed by Agarwal et al. in 2016 to measure the oxidative-reductive potential (ORP) of seminal plasma [71], as it negatively correlated with all standard seminal parameters (sperm concentration, sperm motility, normal morphology and total motile count) [72].

Lastly, a new field of investigation is represented by epigenetics, which plays a pivotal role during spermatogenesis. Both DNA methylation and Histone acetylation can be modified by ROS, inducing not only a low quality of semen, but also damages that can be inherited by the next generation [8]. It is known that epigenetic mechanisms refer to those phenotypic manifestations not related to structural gene modifications but to modulation of their expression, obtained by three main mechanisms, which are DNA methylation, modification of nuclear proteins and metabolism of miRNA. A review on these mechanisms operating in spermatogenesis, all of which are present in the testis, is presented by Sharma [8]. It has been shown that altered DNA methylation is related to seminal abnormalities, such as reduced sperm count, decreased motility and vitality [73]. The histone modifications are crucial, since they influence interactions with DNA and maintain genes in an active or inactive state. The histones undergo different post-translational modifications, such as methylation, phosphorylation, acetylation, sumoylation and ubiquitylation [74]. For instance, acetylation of the histone H4 at lysin 12 is associated with reduced fertility in humans [75]. Moreover, the transition of histones to protamines, as described above, may cause infertility or abnormalities in offspring [76]. Finally, many miRNA are expressed in human testis and influence the spermatogonia differentiation [77]. The pattern of miRNA in patients with seminal abnormalities have been claimed as biomarkers of fertility status [78]. All these mechanisms can be affected by OS [8] with heritable epigenetic changes.

A summary of the main features is schematized in Figure 2.

## 4. Seminal Plasma Antioxidants

Seminal plasma is furnished by several antioxidant defenses, including enzymatic and non-enzymatic molecules, as extensively described in the literature published in the 1990s. As a comparison, we should consider that it has been estimated that the total antioxidant power of seminal plasma is ten times higher than that of blood [79]. Chain-breaking antioxidants have the specific ability to directly trap ROS, preventing amplification of radical formation and the cascade of oxidative damage.

Enzymatic systems comprise superoxide dismutase (SOD), catalase, glutathione peroxidase and reductase, and the systems composed by peroxiredxins and thioredoxin. Non-enzymatic defenses consist of scavenger molecules (i.e., ascorbate, urate, thiol groups), antioxidants that interrupt the propagation of peroxidation (i.e., α-tocopherol) and iron-binding molecules (i.e., transferrin and lactoferrin) [80].

Several studies have tried to establish the origin of redox enzymes, but despite many efforts no conclusive results have been found. Antioxidant systems in rat testicular cells have been studied [81] and a different distribution of SOD and reduced glutathione (GSH) in various cell types of rat testis has been described. In particular, Sertoli and peritubular cells elevated SOD- and GSH-dependent enzymes associated with high GSH content, while pachitene spermatocytes and spermatids showed higher SOD and GSH content, with very low GSH-dependent enzyme activity; finally, spermatozoa had the same enzymatic systems, but were devoid of GSH. Consequently, different cell types displayed variable susceptibility to OS. Lastly, several reports have suggested a post-testicular origin of redox enzymes (with the main contribution of prostatic fluid) to protect ejaculated spermatozoa during the transit in the female reproductive tract [82,83].

However, a major contribution to total antioxidant capacity (TAC) is ascribable to epididymis, which protects spermatozoa during their storage between ejaculations. Accordingly, it has been reported that the seminal plasma of vasectomized men contains lower TAC values, lower thiol groups and a higher amount of lipid peroxidation when compared with intact men [84].

Moreover, low-molecular-weight non-enzymatic scavengers appeared to play a major role in comparison with high-molecular-weight enzymatic molecules [85]. The addition of non-enzymatic antioxidants reduced the amount of lipid peroxidation and DNA damage induced by ROS in vitro [86,87]; an inverse correlation between non-enzymatic antioxidant capacity and lipid peroxidation was described [88,89]; finally, several reports confirmed these data in vivo [88,90], despite the short exposure of spermatozoa to seminal plasma.

Several different techniques have been introduced to evaluate TAC; our group employed the method proposed by Rice-Evans and Miller, with some modifications [91,92]. This method is a kinetic assay based on the ability of antioxidants in seminal plasma to interfere with a reaction between ABTS (2,2′-azino-di-[3-ethyl-benzothiazoline-6-sulfonic acid]) and metmyoglobin after adding H_2_O_2_. The method has also been validated in semen quality assessment and was fairly predictive of antioxidant capacity similarly to the enhanced chemiluminescence assay (a reference method that is unfortunately cumbersome, expensive, and time-consuming) [93]. This method allows one to quantify “Fast” antioxidant capacity (calculated as a lag time in the accumulation of ABTS^●+^ due to the presence of chain-breaking antioxidants) and “total” antioxidant capacity (calculated by absorbance reading at 734 nm after 300 s due to the contribution of fast-reacting and slow-reacting antioxidants). The “Slow” component of antioxidant capacity was calculated as “total” minus “fast” antioxidant capacity (Figure 3).

Antioxidant potential was quantified by using Trolox as a reference standard and expressed as Trolox equivalent in millimolar range (often reported as TEAC values). In our preliminary observations performed in our outpatient clinics for infertility, we observed that in most samples, the “Fast” component of antioxidant potential results were almost equivalent to the “Slow” component, each representing half of antioxidant activity. Interestingly, in several patients, the two components had very different results from each other, the “Fast” or the “Slow” component representing much more than half of antioxidant activity (see in Appendix A). These preliminary data are very encouraging and appear more informative than the conventional approach regarding a single component alone of semen antioxidant potential to evaluate the involvement of oxidative insult in the etiology of human fertility disorders. Therefore, we suggest that the “Fast” and “Slow” antioxidant capacity of seminal plasma should be included as an informative parameter indicative of reproductive health status.

Interestingly, we also found that seminal plasma TAC can be modulated also by the systemic hormonal milieu. In a study performed in a group of infertile patients we found an inverse correlation between prolactin (PRL) and sperm motility, and a direct correlation of TAC with PRL and free thyroxine (fT4), but not with gonadotropins or gonadal steroids. It must be underlined that such hormones are not usually tested in first-level evaluation of male patients with fertility problems [94] and that, since thyroid hormones play a pivotal role in antioxidant modulation [95], a relationship could exist between blood and the seminal compartment under this profile.

A special role is played by Coenzyme Q10 (CoQ), a component of the mitochondrial respiratory chain but also a potent antioxidant, due to its recycling between oxidized/reduced forms and interacting with other antioxidants such as Vitamin E [96]. After an empirical original report on its effect, our group was the first that reported its determination in total semen and in seminal plasma in humans [97]. The CoQ concentrations directly correlated with spermatozoa concentration and % of motility. An interesting exception was represented by varicocele, which maintained the correlation between CoQ and sperm count, but not with sperm motility.

Moreover, varicocele subjects exhibited higher plasma CoQ than intracellular, suggesting an altered utilization by nemaspermic cells [98]. A similar pattern was also reported for another biomarker of inflammation in seminal plasma (e.g., free light chain k), underlying the link between OS and inflammation [99].

In idiopathic oligozoospermic patients, intracellular CoQ was inversely correlated with sperm count; in this case, a possible compensatory mechanism ensues [100].

Moreover, TAC values were increased in the seminal plasma of varicocele patients. Interestingly, they were influenced by different medical therapies used for male infertility, such as CoQ supplementation [101]; the last point underlines the functional interconnection between antioxidants. In addition, FSH injection, other than stimulating spermatogenesis, can have a positive effect on antioxidant defenses. Finally, in the model of hypo-gonadal patients as cited above, testosterone administration [102] induced an increase in TAC values (Table 1).

Interestingly, surgical treatment of varicocele influences both CoQ10 and TAC. In fact, in a previous paper, we reported that LAG values were higher in VAR patients vs. controls (reaching significance in the group of oligospermic patients) and apparently were not modified by intervention; however, while before surgery TAC significantly correlated with sperm motility, this correlation was lost after surgical treatment. Thus, we hypothesize that surgery can induce and re-establish an equilibrium in the protective role of antioxidants toward sperm motility [103].

## 5. Conclusions

The dual role of ROS is increasingly becoming more evident. Due to the physiological role of ROS, which is crucial for maturation, motility and the ability to fertilize, and the role of OX in male infertility, several trials have been conducted with the supplementation of natural or pharmacological antioxidants [104,105,106,107], but there is no definitive demonstration of their efficacy [108,109]. On the contrary, an indiscriminate administration of antioxidants could induce a deterioration of seminal picture and therefore reduce, instead of ameliorate, male fertility potential. The concept of “reductive stress” describes such a situation well. Therefore, despite the determination of ROS and products of OS not being recommended in previous WHO guidelines [110], the evaluation of the redox state of a sample (i.e., seminal plasma) should be performed before establishing a treatment, and a panel of biomarkers with specific goals is recommended in semen quality evaluation. Here, we suggest that the “Fast” and “Slow” antioxidant capacity of seminal plasma should be included as an attractive biomarker indicative of the reproductive health status.

Finally, OS can be the only recognized cause of unexplained infertility, providing another reason to evaluate it. The identification of appropriate and easy-to-determine biomarkers of such situations still requires future investigation.

Another clinical important aspect is related to assisted reproduction, since some methods of sperm selection can induce sperm damage via oxidative stress, as extensively discussed [111]. As previously stated, the importance of lifestyle factors should be highlighted, since they are under-evaluated in the clinical approach to infertile couples [112]. Moreover, a better identification of OS and a more extensive application of simple methods, such as TAC, could help to bridge the gap that still exists between basic studies and clinical applications. Clinical trials evaluating specific dietetic or pharmacological treatments should determine if sperm amelioration is effectively linked to the improvement of oxidative status.

Researchers should discuss the results and how they can be interpreted from the perspective of previous studies and of the working hypotheses. The findings and their implications should be discussed in the broadest context possible, highlighting future research directions.

## Figures and Tables

**Figure 1 ijms-24-04994-f001:**
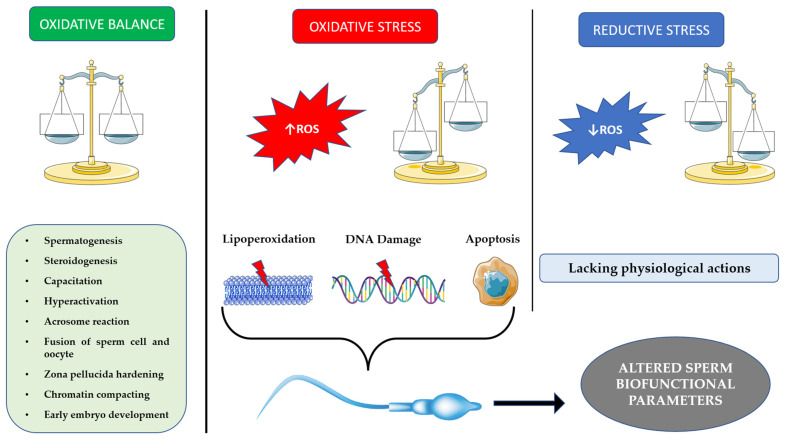
Left panel: redox homeostasis with positive effects of ROS; central panel: oxidative stress situation due to increased ROS production and consequential oxidative damages; right panel: reductive stress situation with low ROS levels and consequential detrimental effects.

**Figure 2 ijms-24-04994-f002:**
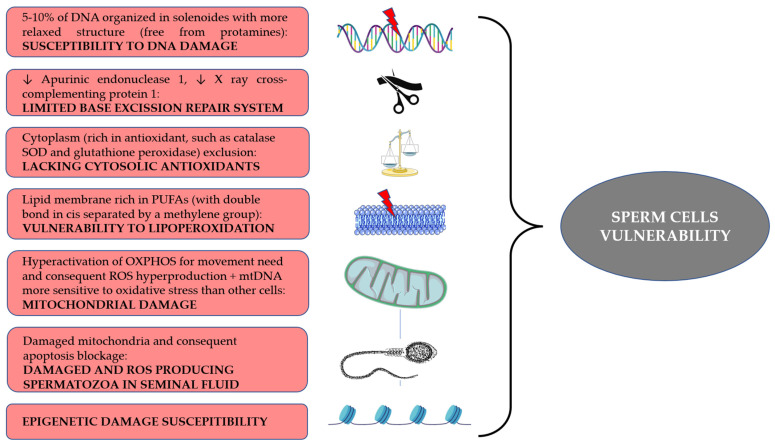
Infographic summary of main characteristics of spermatozoa associated with their vulnerability toward ROS attack.

**Figure 3 ijms-24-04994-f003:**
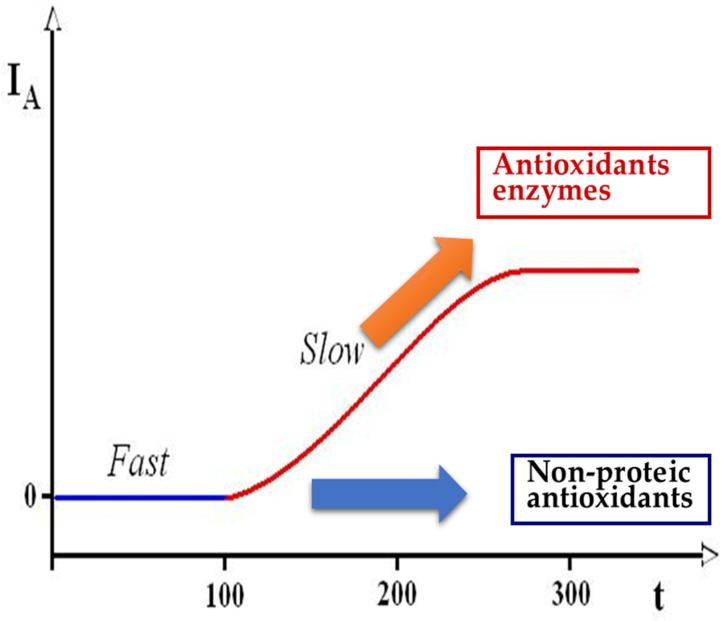
The components of “total” antioxidant capacity of seminal plasma, i.e., “Fast” (low-molecular-weight chain-breaking antioxidants) and “Slow” (antioxidant enzymes, e.g., SOD). In ordinate, “I_A_” represents the absorbance at 734 nm; in abscissa, “t” represents the time expressed in seconds. ABTS radical species appear after the initial period (latency phase) with the enrollment of “Fast” component, then gradually increase until a plateau related to the “Slow” component.

**Table 1 ijms-24-04994-t001:** Effects of medical and surgical treatments on TAC values (in seconds) in varicocele patients, classified according to seminal analyses (oligozoospermic and normozoospermic subjects).

	CoQ10 (200 mg/die)	FSH (75 U i.m. Every Two Days)	Testosterone (250 mg Every Three Weeks)	Surgery (Oligo)	Surgery (Normo)
TAC (s) before treatment	106 ± 8.7	71.7 ± 11.4	67 ± 4.9	126.6 ± 19.7	95.3 ± 7.4
TAC (s) after treatment	148 ± 12.9 ^1^	90.0 ± 15.2	78.3 ± 34.6 ^1^	102.5 ± 15.4	108.1 ± 15.4

(^1^
*p* < 0.05 vs. values before treatment).

## Data Availability

Not applicable.

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
