# Peer review of "The Dual Role of Oxidants in Male (In)fertility: Every ROSe Has a Thorn"

_ijms, 2023, doi:10.3390/ijms24054994_

Round 1
Reviewer 1 Report
The authors reviews the roles of oxidative stress in male infertility. However, I didn't see much novel information, and most the information has been covered by recent reviews, such as Evans et al. (2021), and Barati et al. (2020). Therefore, I don't think it can meet the standards of IJMS.
Author Response
Rev1
The authors reviews the roles of oxidative stress in male infertility. However, I didn't see much novel information, and most the information has been covered by recent reviews, such as Evans et al. (2021), and Barati et al. (2020). Therefore, I don't think it can meet the standards of IJMS.
Thanks for work done and for your comment. We agree that some papers have been published about the topic, but it is still very hot in clinical practice, due to excessive and inappropriate use of antioxidant preparations. Therefore, our focus was directed to physiological ROS role, more specifically than the two reviews indicated, which have been added in the discussion.
Moreover, we have improved the overall manuscript also adding new infographic. Surprisingly, supplementation with antioxidants is still prescribed without a precise clinical picture, and the question still remains open, therefore we believe that focusing attention on this point also deepening the clinical part can improve the discussion of this aspect allowing further participation of the scientific community.
Reviewer 2 Report
See in attachment

Author Response
Rev2
The review is well written and very interesting. It only requires the addition of some information and the editing of some parts of the manuscript. The work can be accepted with minor changes
Thanks for this positive comments and for your suggestions.
MINOR REVISION
Figure
- The authors should improve the quality of Figure 1. The figure is also unclear at first glance, not readable, the authors should make it better, and make it more appealing.
We have ameliorated the Fig 1 (now figure 3) also adding some explanation in the legend.
- The authors could add the summary infographic for a summary of information, to allow readers a better overall understanding of the work
Infographic figures have been added to summarize main information in chapters 2 and 3.
General informations:
- The authors should improve the paragraph of review:
- such as the paragraph on sources of ROS (example immune system, immature spermatozoa, etc.)
- paragraph on ROS techniques identification and add information regarding MiOXISIS system for evaluation of ROS in the semen
- Paragraph on how ROS levels can be used as a marker of seminal fluid quality
Thank you. We add some sentences to improve all these aspects. Paragraphs on sources of ROS and assay methods (including MiOXISIS) have been added in chapter 2, while the discussion on role of TAC as biomarker has been extended.
Reviewer 3 Report
This manuscript provides a comprehensive review of oxidants in male fertility. Highlighting the dual aspect of reactive oxygen species (ROS) during sperm capacitation, hyperactivation, acrosome reaction, and oocyte fusion. The manuscript is clearly written, well documented, and references are relevant. In addition, the work is well presented and written in a logical way. I would like to make some suggestions to help improve the manuscript.
Title:
- - I find the title witty but a bit informal for a scientific article. It could be used in oral communications but for the publication of a scientific paper I consider that it would have to be more rigorous. So, I propose to remove the first part of the title and use the second part: The Dual Role of Oxidants in Male Infertility
Abstract:
- - Line 12: add a comma after However
Introduction:
- - In line 37 three phases are named but, in the text, only the second (line 44) and third (line 45) are specified. Add which part of the text corresponds to the first phase.
- - Line 45: add a comma after Finally
Positive effects of ROS:
- - In general, I consider that the article is very well written and documented, but the visual iconography could be improved, since there is only one figure throughout the text. I propose that the authors add an illustrative figure to show the importance of the involvement of ROS during sperm capacitation, hyperactivation, acrosome reaction, and oocyte fusion. Or some illustration that summarizes the information in the text.
In the following links you can find examples of figures related to ROS and sperm:
https://www.mdpi.com/2076-3921/11/2/264
https://www.frontiersin.org/articles/10.3389/frph.2022.822257/full
https://www.ncbi.nlm.nih.gov/pmc/articles/PMC5456340/
- - Line 133: replace cumuli by cumulus
- - Line 135: Since each species has a specific oocyte receptor or receptors, I suggest specifying that in human sperm, an important role is given to the interaction with the ZP3 protein of the oocyte.
- - Line 140: replace acrosomial by acrosomal or acrosome
Seminal Plasma Antioxidants:
- - I have not been able to locate the citation for Figure 1 in the text. Review this and if it is not, add the citation in the text. Also, add absorbance units on the ordinate axis.
- - Add the units to table 1
- - Line 345: add a comma after Here
- - Line 346: What do you mean by unselected patients?
- - Line 344-361: It would be interesting to know the data on the oxidative capacity in normozoospermic controls. Have you carried out controls on the antioxidant capacity of normozoospermic donors without infertility disorders? In addition, the term infertility disorders (line 347) is very broad. You should limit the study and introduce inclusion and exclusion parameters in order to have more concrete results. In case there are no controls or inclusion criteria, I recommend eliminating this part of the manuscript since they are very preliminary results.
Author Response
Rev3
This manuscript provides a comprehensive review of oxidants in male fertility. Highlighting the dual aspect of reactive oxygen species (ROS) during sperm capacitation, hyperactivation, acrosome reaction, and oocyte fusion. The manuscript is clearly written, well documented, and references are relevant. In addition, the work is well presented and written in a logical way. I would like to make some suggestions to help improve the manuscript.
Thank you for your positive comments
Title: - I find the title witty but a bit informal for a scientific article. It could be used in oral communications but for the publication of a scientific paper I consider that it would have to be more rigorous. So, I propose to remove the first part of the title and use the second part: The Dual Role of Oxidants in Male Infertility
Thanks. We have changed the title according to your suggestions. However, we prefer to leave as subtitle the common sentence to be more attractive.
Abstract:
- Line 12: add a comma after However
Done
Introduction:
- In line 37 three phases are named but, in the text, only the second (line 44) and third (line 45) are specified. Add which part of the text corresponds to the first phase.
We have specified the reference corresponding to the text
- Line 45: add a comma after Finally
We have done all these changes, thank you!
Positive effects of ROS:
- In general, I consider that the article is very well written and documented, but the visual iconography could be improved, since there is only one figure throughout the text. I propose that the authors add an illustrative figure to show the importance of the involvement of ROS during sperm capacitation, hyperactivation, acrosome reaction, and oocyte fusion. Or some illustration that summarizes the information in the text.
In the following links you can find examples of figures related to ROS and sperm:
https://www.mdpi.com/2076-3921/11/2/264
https://www.frontiersin.org/articles/10.3389/frph.2022.822257/full
https://www.ncbi.nlm.nih.gov/pmc/articles/PMC5456340/
Thank you for all these constructive comments. We have added an infographic figure (for chapters 2 and 3) to show the importance of the involvement of ROS during sperm capacitation to improve this aspect.
- Line 133: replace cumuli by cumulus
Done
- Line 135: Since each species has a specific oocyte receptor or receptors, I suggest specifying that in human sperm, an important role is given to the interaction with the ZP3 protein of the oocyte.
We have specified “in humans”
- Line 140: replace acrosomial by acrosomal or acrosome
Done
Seminal Plasma Antioxidants:
- I have not been able to locate the citation for Figure 1 in the text. Review this and if it is not, add the citation in the text. Also, add absorbance units on the ordinate axis.
We have cited the figure in the text and improved the figure legend.
- Add the units to table 1
Done
- Line 345: add a comma after Here
Done
- Line 346: What do you mean by unselected patients?
The term “unselected” and “infertility disorders” have been deleted; we have also deleted the most part of presentation of our preliminary results and shortened the considerations about the fast and slow component of TAC we are investigating.
- Line 344-361: It would be interesting to know the data on the oxidative capacity in normozoospermic controls. Have you carried out controls on the antioxidant capacity of normozoospermic donors without infertility disorders? In addition, the term infertility disorders (line 347) is very broad. You should limit the study and introduce inclusion and exclusion parameters in order to have more concrete results. In case there are no controls or inclusion criteria, I recommend eliminating this part of the manuscript since they are very preliminary results.
According to your comments we decide to reduce drammatically this part.
Reviewer 4 Report
Mancini et al. have nicely compiled a lot of information about the role of oxidants in male fertility. It is a nice compilation of different publications covering different aspects of oxidative stress and its role in male fertility.
There are some aspects that have not been well elaborated and there are some missing citations that should be added. Line wise comments can be seen below:
Line 41 - discover should be changed to discovery
Line 50-52 - Authors are discussing about the involvement of ROS in cytokine and growth factor signaling but some of the prominent and recent publications are not cited (such as https://pubmed.ncbi.nlm.nih.gov/25402766/ & https://www.ncbi.nlm.nih.gov/pmc/articles/PMC7481202/). They should also describe briefly how ROSs are typically involved in these conditions.
line 105 - it should be "consists of a series"
line 131-132 - in this paragraph, cite more recent papers such as (https://www.ncbi.nlm.nih.gov/pmc/articles/PMC4016371/#:~:text=Superoxide%20dismutase%20(SOD)%20is%20an,protecting%20spermatozoa%20from%20oxidative%20damage.) Also add some description about the recent research.
lines 133-147 - check the spelling of "serine"
Line 164-167 - rephrase the sentence. It is not written in an understandable way
Line 173 - "Normally ROS damage...." - this statement is missing a citation. Provide with a citation.
Line 204 - "as the second step......" - rephrase this statement.
Line 219 - authors mentioned that peroxiredoxins play an important role as well. It would be important to elaborate a little bit more on that.
Line 265 - "Despite mtDNA ....." - the statement is grammatically incorrect. Rephrase it.
Line 269- " All of these factors..." - rephrase it to make it more clear what the authors want to say.
line 284 - Authors have mentioned about epigenetics which is an interesting aspect. Can you provide with 2-3 more sentences for elaborating it.
line 296 - enzymatic systems would also include peroxiredoxins and thioredoxin system. also provide citations in this paragraph.
line 312 - "However, a major contribution....." - elaborate it and add more recent citations (like https://www.ncbi.nlm.nih.gov/pmc/articles/PMC3708257/)
Supplementary figures are missing the captions about what the different colors represent.
Author Response
Rev4
Mancini et al. have nicely compiled a lot of information about the role of oxidants in male fertility. It is a nice compilation of different publications covering different aspects of oxidative stress and its role in male fertility.
Thank you for your positive comments.
There are some aspects that have not been well elaborated and there are some missing citations that should be added. Line wise comments can be seen below:
Line 41 - discover should be changed to discovery
Done
Line 50-52 - Authors are discussing about the involvement of ROS in cytokine and growth factor signaling but some of the prominent and recent publications are not cited (such as https://pubmed.ncbi.nlm.nih.gov/25402766/ & https://www.ncbi.nlm.nih.gov/pmc/articles/PMC7481202/). They should also describe briefly how ROSs are typically involved in these conditions.
We have added studies about peroxiredoxins
line 105 - it should be "consists of a series"
Done
line 131-132 - in this paragraph, cite more recent papers such as (https://www.ncbi.nlm.nih.gov/pmc/articles/PMC4016371/#:~:text=Superoxide%20dismutase%20(SOD)%20is%20an,protecting%20spermatozoa%20from%20oxidative%20damage.) Also add some description about the recent research.
We have added some sentences about recent data on SOD
lines 133-147 - check the spelling of "serine"
Done
Line 164-167 - rephrase the sentence. It is not written in an understandable way
The sentence has been rephrased.
Line 173 - "Normally ROS damage...." - this statement is missing a citation. Provide with a citation.
The citation has been inserted.
Line 204 - "as the second step......" - rephrase this statement.
The sentence has been rephrased.
Line 219 - authors mentioned that peroxiredoxins play an important role as well. It would be important to elaborate a little bit more on that.
The discussion about peroxiredoxins has been extended.
Line 265 - "Despite mtDNA ....." - the statement is grammatically incorrect. Rephrase it.
The sentence has been corrected and rephrased.
Line 269- " All of these factors..." - rephrase it to make it more clear what the authors want to say.
The sentence has been corrected and rephrased.
line 284 - Authors have mentioned about epigenetics which is an interesting aspect. Can you provide with 2-3 more sentences for elaborating it.
We have added a paragraph about epigenetic mechanisms involved in infertility
line 296 - enzymatic systems would also include peroxiredoxins and thioredoxin system. also provide citations in this paragraph.
The two systems have been added
line 312 - "However, a major contribution....." - elaborate it and add more recent citations (like https://www.ncbi.nlm.nih.gov/pmc/articles/PMC3708257/)
We have extended the discussion with more recent papers.
Supplementary figures are missing the captions about what the different colors represent.
We have indicated the meaning of the colors in the supplementary figures.
Round 2
Reviewer 1 Report
I still think the novelty of this manuscript is below my expection. However, I would respect the decision of the editor and other reviewers.
Reviewer 3 Report
The authors have addressed the concerns and incorporated them into the manuscript.